



# Dynamics of finite causal processes

Kalman Ziha

University of Zagreb, Faculty of Mechanical Engineering and Naval Architecture, 10002 Zagreb, Ivana Lucica 5, Croatia

*Correspondence to*: Kalman Ziha (kziha@fsb.hr)

**Abstract.** Earth and environmental mechanisms and phenomena are often physically finite dynamical causal processes and need more precise mathematical elaboration. Therefore this article at the beginning resumes the decomposition of general infinite circular causal relations with linear feedbacks to primary causal effects and to interactions among boundless effects and causes. In the sequel it reveals the mathematical model of general finite cause-and-effect interaction with non-linear feedback induced by finiteness of causal processes with exhaustible causal capacities. The study also uncovers that the reverse

application of the mathematical model makes it possible to discover and to estimate the unknown ultimate causal capacities from relevant information of supposedly finite causal processes beyond the instant of observation. The article at the end demonstrates that the environmental relations among global climate change and ice mass losses monitored recently on Greenland and Antarctica ice sheets are plausibly finite dynamical climate processes in interaction with cryosphere.

## 1 Introduction

This study aims to find out if there are synthetic measurable theoretically founded quantitative parameters of a general mathematical model appropriate for analysis and predictions of complex interactions of natural finite causal processes. The mathematical model of finite dynamical causal processes in the article follows the concepts of quantitative theory of growth (von Bertalanffy, 1938), control systems (Wiener, 1948) and open systems in general system theory (von Bertalanffy, 1968). Natural Cause-and-Effect relations (CE) and Circular Causation interrelations (CC) of interacting Effects ($E$) and Causes ($C$)

are in reality often limited by available ultimate normally exhaustible and finite causal capacities $C_U$. The empirical problem of interactions of limited causal processes is tackled in the study by a general mathematical model of Finite Cause-and-Effect Interaction (FCEI) concept. The guiding thought is that complex interactions of finite natural causal processes and limited environmental resources under multifarious conditions implying feedbacks ($F$) and interactions ($I$) can be mathematically decomposed into simple CE and to finite FCEI relations. The study embraces that the mere existence of finiteness of causal

processes itself induces non-linear feedbacks $F[E(C)]$ of effects $E(C)$ of causes $C$ in the FCEI processes. These feedbacks are the results solely of the fact that there are real limited residual driving causes $R=C_U-C$ after some cause $C$ elapsed on the expense of the ultimate causal capacity $C_U$. The finite interaction $I(E,C)$ between the effect $E$ and the cause $C$ is regarded in this study as the forthcoming effect of the feedback $F[E(C)]$ to effect $E(C)$ induced only by the finite causal capacity $C_U$. Accordingly, the FCEI concept may be applied in reverse direction in discovery and estimation of unknown ultimate capacity



$C_U$ from past partial data of supposedly finite causal processes in the future. The FCEI concept basically represent the physical interchanging processes of physical entities among elapsed effects and forthcoming residual causes beyond the moments of direct observations. The concept was tested earlier on problems of fatigue yielding (Ziha 2009), general considerations on how things worsen (Ziha 2012), ageing and fatigue (Ziha 2014), material plasticity (Ziha, 2015) and fatigue life predictions in engineering (Ziha 2016). The study investigates and demonstrates the appropriateness of the mathematical model of Finite ice Melting (the cause $M$) and ice mass Losses (the effect $L$) Interaction (FMLI) on the numerical analysis of the interaction of climate changes and the recently observed alarming land ice mass anomaly of Greenland and Antarctica ice sheets (Velicogna and Wahr, 2006a; Velicogna and Wahr, 2006b; Velicogna, 2009; Sasgen et al., 2013; Velicogna et al., 2014;; Tedesco et al., 2015; Tedesco et al., 2016; Tedesco et al. 2017; Wiese et al., 2016).

## 2 Concept of general cause and effect interaction

Traditional readings on causation in terms of invariable patterns of succession lead to regularity theories which imply that the cause $C$ and the effect $E$ are connected but different entities. That $C$ affects $E$ is the singular causal claim $C{\Rightarrow}E$ where $C$ and $E$ are relata of the claim. The prevailing present comprehension of causality is that the knowledge of causal relations arises entirely from experience. The conclusions made from experience imply a belief that some observable courses of nature could be sufficiently uniform so that the future would be conformable to the past. There is no assertion with respect to directionality of causation, except for the common experience that an effect does not influence in reverse the cause $C \not\Leftarrow E$ (asymmetry). It is commonly agreed that the flow of causality proceeds from past to the present into the future which are unavoidably separated by the inherent human ability of perceiving only at the instant of observation. Such a belief makes possible the prediction of the course of progress of a CE relation beyond the moment of observation unnefected by the future. In physical view of causal determinism the world-at-a-time has an objective notion in which the particular causes $C$ and effects $E(C)$ of $C$ are normally regarded as empirical laws of nature. The claim, according to which every later effect is uniquely determined by its earlier cause, doesn't necessarily regards interactions of $E$ and $C$.

### 2.1 The simple infinite cause and effect mathematical relation

The simple CE relation of a causal model where the unaffected primary cause is directly linearly applied to the primary effect $E'(C)$ in proportion $p$ that represents the CE progression factor, can be mathematically presented as:

$$E'(C) = p \cdot C \qquad (1)$$

An infinite causal relation is theoretically not limited in its progression and may unaffectedly continue beyond the instant of observation following the empirical causal term in (1). The rate of change of the primary infinite causal relation (1) in which the ultimate causal capacity $C_U$ is undefined, i.e. considered infinite, is simply constant:

$$\mathrm{d}E'(C)/\mathrm{d}C = E'(C)/C = p \qquad (2)$$





## 2.2 The infinite cause and effect interaction induced by feedback

The general infinite CE relation in (1) and (2) is the linear CE relation representing an open loop system in system dynamics where the parameter $p$ is denoted as the open-loop gain. However, the general CE relation (1) in open and active environment can ensue as a continuous relation in which the elapsed effect $E'(C)$ induced by the primary cause $C$ at the instant of observation

has been influencing the forthcoming secondary cause $C''$ (Wiener, 1948; von Bertalanffy 1968). The fraction $f$ of the primary effect $E'(C)$ in (3) may turn to feedback $F$ affecting the primary cause $C$ in the C-E space where $f$ is denoted in the control theory as the feedback factor. The feedback $F$ to primary effect $E'(C)$ is defined as:

$$F\left[E'(C)\right] = f \cdot E'(C) = f \cdot p \cdot C \tag{3}$$

The definition of the secondary cause $C''\{F[E'(C)]\}$ induced by the feedback $F[E'(C)]$ (3) to the primary effect $E'(C)$ (1) (Fig.

1) which is affecting in turn the primary cause $C$ is then defined as follows:

$$C'' = C + F\left[E'(C)\right] = C + f \cdot E'(C) = C + f \cdot p \cdot C = (1+i) \cdot C \tag{4}$$

In (3) and (4) $i = f\,p$ is the parameter combined of the feedback factor $f$ in (3) and progression factor $p$ in (1) that represents the intensity of interaction between the elapsed effect and the forthcoming cause.

Substituting the value of the primary cause $C$ from (4) into the primary effect (1) it follows

$$E'(C) = p \cdot \left\{C'' - F\left[E'(C)\right]\right\}.$$

The primary effect $E'(C)$ is after rearrangement expressed by the forthcoming cause $C''$ (Fig. 1) as shown:

$$E'(C) = C'' \cdot p / (i+1) \tag{5}$$

Terms (3) and (4) represent the circular causality (CC) of interrelated sequence of interacting cause $C$ and effect $E$ with constant feedback $F$ (3) that form a closed-loop system like a circuit or a loop.

## 2.3 The mathematical decomposition of the causal relation

In some temporal circular causal (CC) systems the delivery of the feedback $F$ in (3) may occur instantaneously or in a very short time unimportant for outcome of the CE relation for a conscious observer. The overall effect $E(C)$ of the cause $C$ at the perceptible moment of observation may be then taken as equal to the effect of the CE relation (1) up to the forthcoming effect $C''$ induced by the feedback $F$, that is, $E'(C'')=p\,C''$ (Fig. 1). The overall effect $E(C)$ at the moment of observation is

mathematically decomposable to the primary elapsed effect $E'(C)$ as in (1) and to the effect of feedback $E'\{F[E'(C)]\}=p\,F[E'(C)]$ as in (3) for the whole range of the cause $C$ as shown:

$$E(C) = E'(C'') = p \cdot C'' = (1+i) \cdot E'(C) = E'(C) + E'\left\{F\left[E'(C)\right]\right\} = E'(C) + I(E,C) \tag{6}$$

The effectiveness $q$ of the circular causal relation with feedback (5) is denoted as the closed-loop gain and can be presented as

$$q = E'(C) / C'' = E'(C) / E(C) / p = p / (1+i).$$





The temporal effect of feedback $E'\{F[E'(C)]\}$ in (6) is defined in this study as the interaction $I(E,C)$ at the moment of observation between the forthcoming cause $C''$ in the future and the elapsed primary effect $E'(C)$ in the past (Fig. 1). The interaction $I(E,C)$ depending on the intensity $i$ representing both the feedback factor $f$ and the open-loop gain $p$, is then:

$$I(E,C) = E'\{F[E'(C)]\} = p \cdot F[E'(C)] = E'[f \cdot E'(C)] = f \cdot p \cdot E'(C) = i \cdot p \cdot C \tag{7}$$

5  Note that the feedback in (3) is inversely related to the interaction (7) as $\{F[E'(C)]\} = E'^{-1}[I(E,C)]$.

The rate of change of the interaction term for constant feedback is simply constant obtained by chain derivative of (7):

$$\frac{\mathrm{d}I(E,C)}{\mathrm{d}C} = \frac{E'\{F[E'(C)]\}}{C} = \frac{\mathrm{d}I(E,C)}{\mathrm{d}F(E,C)} \cdot \frac{\mathrm{d}F(E,C)}{\mathrm{d}E'(C)} \cdot \frac{\mathrm{d}E'(C)}{\mathrm{d}C} = f \cdot p^2 = i \cdot p \tag{8}$$

Note also that for $E'(C)=C$ is $p=1$ in (1), and consequently, the interaction (7) is equal to the feedback (3), that is $I(E,C)=\{F[E'(C)]\}$. The appropriate rate of change of the overall effect $E(C)$ (6) with feedback (3) (Fig. 1) is

10  $\mathrm{d}E(C)/\mathrm{d}C = p \cdot (1+i)$.

The overall theoretical potential $W$ of interaction $I(E,C)$ (7) can be calculated by the integration of (7) up to the primary cause $C$ at the instant of observation as follows:

$$W[I(E,C)] = \int_0^C I(E,C)\,\mathrm{d}C = i \cdot E'(C)^2 / (2 \cdot p) = i \cdot p \cdot C^2 / 2 \tag{9}$$

The interaction intensity parameter $i$ in (7) and (8) can be calculated directly from the known interaction potential $W$ (9):

15  $i = 2 \cdot p \cdot W[I(E,C)] / E'(C)^2 = 2 \cdot W[I(E,C)] / (p \cdot C^2)$  (10)

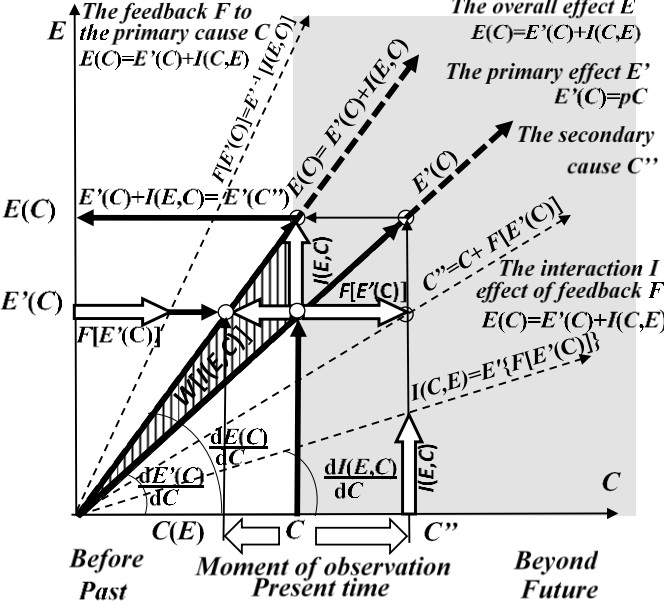

**Figure 1: Decomposition of general cause-and-effect relation with linear feedback in the C-E space.**



## 3 Finite cause and effect interaction

This study of finiteness of natural processes recognises the temporal FCEI empirical concept as a continuous sharing of irreplaceable and restricted overall ultimate causal capacity $C_U$ between the observable elapsed effect $E$ in the past and the imperceptible but conceivable forthcoming limited exhaustible cause $C$ beyond the instant of observation in the future (Fig.

2). The trans-temporal finite interaction implies the empirical link in continuation of the known uninterrupted past and the imaginable but finite perpetuating future separated by the instant of observation at the present time. The finiteness affects causal processes solely by the fact that there is really a limited ultimate and final causal capacity $C_U$ (see Appendix A).

The mathematical model of the FCEI in this study considers a simple intuitive term of the continuous residual causal capacity $R(C)$ after spending some primary effect $E'(C)$ (1) of the limited cause $C$ on the expense of the ultimate cause $C_U$. The leftover

driving cause $R(C)$ for the future uninterrupted perpetuation of the CE relation is defined in a linear form as:

$$R(C) = (C_U - C) = C_U \cdot (1 - c) \tag{11}$$

In (11) $c = C/C_U$ is the relative linear progression of the elapsed cause $C$ with respect to the ultimate causal capacity $C_U$.

### 3.1. The feedback of finiteness to causal processes

The following expression defines the mathematical rate of the finite interaction $dI_F(E,C)$ as the relation of the effect

$E'\{F[E'(C)]\}$ of the feedback $F[E'(C)]$ (3) of the elapsed effect $E'(C)$ (1) and the remaining capacity $(C_U-C)$ (11) instead of the mere effect of the progressing linear cause $C$ as it is in the general infinite interaction model (8) (Fig. 2) as shown:

$$\frac{dI_F(E,C)}{dC} = \frac{E'\{F[E'(C)]\}}{R(C)} = \frac{dE'(C)}{d(C_U - C)} = i \cdot p \cdot \frac{C}{(C_U - C)} = \frac{dI(E,C)}{dC} \cdot \frac{c}{1-c} \tag{12}$$

The accumulation of the forthcoming finite interactions depends on the remaining cause $R(C)$ (11) rather than on the elapsed effect $E'(C)$ as in (8) and can be calculated by integration of the differential equation of the variable rate of change (12) until

the primary cause $C$ at the instant of observation (see Appendix A) as follows:

$$I_F(E,C) = \int_0^C \frac{dI_F(E,C)}{dC} dC = -i \cdot p \cdot C + i \cdot p \cdot \left( \frac{C_U}{C} \cdot \ln \frac{C_U}{C_U - C} \right) \cdot C = i \cdot p \cdot u(c) \cdot C = I(E,C) \cdot u(c) \tag{13}$$

In (13) $u(c)$ is the dimensionless finite interaction intensity correction function of the general interaction term (7) for limited causal capacity (11) of logarithmic shape and of asymptotic character depending only on the relative FCEI progression $c$ as:

$$u(c) = -1 + \frac{1}{c} \cdot \ln \frac{1}{1-c} \tag{14}$$

The variable rate of sharing of the causal capacity $C/(C_U-C)$ at the moment of observation between the elapsed cause $C$ in the past and the remaining cause $(C_U-C)$ (12) in the future characterizes the influence of the causal finiteness on the progression of the causal process. The overall effect $E(C)$ is then a FCEI where the finite interaction $I_F(E,C)$ (13) analogously to the general interaction $I(E,C)$ in (6) influences the primary effect $E'(C)$ (1) (Fig. 2) as shown:



$$E(C) = E'(C) + I_F(E,C) = p \cdot C + p \cdot F[E'(C)] = p \cdot C + i \cdot p \cdot u(c) \cdot C \qquad (15)$$

The interaction rate (12) geometrically represents the continuously changing angles of tangents on the interaction curve (13) which determine the dynamics of the progression of the FCEI relation (Fig. 2). The effectiveness $q$ of the circular causal relation with feedback (15) is denoted as the closed-loop gain $q = E'(C)/E(C)/p = p/[1 + i \cdot u(c)]$.

The ending cause $C_E$ of the ultimately attainable effect $E(C_E)$ (15) (Fig. 2) is the solution of the inverse of equation of overall effect (15) as shown:

$$C_E = c_E \cdot C_U = E^{-1}[E'(C = C_U)] \qquad (16)$$

The ending cause $C_E$ in (16) is not algebraically separable from (15).

The overall finite interaction potential $W_F$ can be calculated by the integration of (13) (Fig. 2), as shown below:

$$W_F[I_F(E,C)] = \int_0^C I_F(E,C)\,\mathrm{d}C = C_U \cdot i \cdot p \cdot w(c) \cdot C \qquad (17)$$

In (17), $w(c)$ is the dimensionless interaction potential function depending only on the relative progression $c$ as follows:

$$w(c) = -c/2 - u(c) - \ln(1-c) \qquad (18)$$

The interaction intensity parameter $i$ can be calculated from the definition of the overall interaction potential $W_F$ (17) as:

$$i = \{W_F[I_F(E,C)]\} / [p \cdot C_U \cdot w(c) \cdot C] \qquad (19)$$

The second derivative of the overall effect $E(C)$ (15), i.e. the sensitivity of the interaction rate (12) is as follows:

$$\frac{\mathrm{d}^2 E(C)}{\mathrm{d}C^2} = \frac{i}{C_U} \cdot p \cdot \frac{1}{(1-c)^2} \qquad (20)$$

The mathematical definition of the derivatives of functions with respect to bounds of finite variables (Fig. 2) is given in Appendix A.

The variable term $F[E'(C)]$ for feedback is a direct consequence of the effect of finiteness. It is inversely related to the
interaction term (13) (Fig. 2) as $F[E'(C)] = E'^{-1}[I_F(E,C)] = I_F(E,C)/p = i \cdot u(c) \cdot C$.

The secondary cause $C''\{F[E'(C)]\}$ (4) induced by the variable feedback $F[E'(C)]$ to the primary effect $E'(C)$ (Fig. 2) which is affecting the finite primary cause $C$ is then as $C'' = C + F[E'(C)] = C + i \cdot u(c) \cdot C$.

The effectiveness of the FCEI relation (15) is $q = E'(C)/C'' = p/[1 + i \cdot u(c)]$ with respect to the closed-loop gain in (5).

The separation of the cause $C(E)$ from the nonlinear part of the FCEI (13) $I_N(C) = i \cdot p \cdot C_U \cdot \ln[C_U/(C_U - C)]$ corresponds to the
von Bertalanffy asymptotic growth function (von Bertalanffy, 1938) (VBGF) of the cause $C$ in the form $C[I_N(C)] = C_U \cdot [1 - e^{-k \cdot I_N(C)}]$ where $k = 1/(ipC_U)$ and $C_U$ represents the ultimate growth of the cause $C$ depending on the effect $E$.



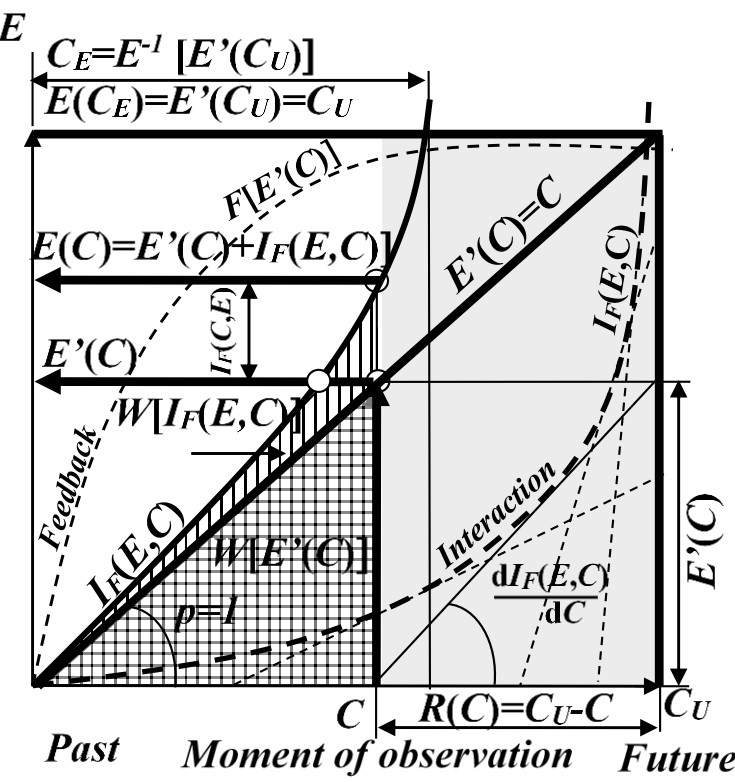

**Figure 2: Decomposition of finite cause and effect interaction in C-E space.**

### 3.2. Numerical estimation of the ultimate causal capacity $C_U$

5  The FCEI mathematical model (11-20) normally provides the interaction term $I_F$ (13), the overall effect $E(C)$ (15), the rate of change $dE/dC$ (12), the sensitivity $d^2E/dC^2$ (20), the theoretical interaction potential $W_F$ (17) and interaction intensity $i$ (19) of finite processes for known ultimate causal capacity $C_U$.

The reverse FCEI mathematical procedure makes it possible to estimate the unknown ultimate causal capacity $C_U$ and the interaction intensity $i$ from the past data of supposedly finite causal processes even beyond the instant of observation.

The numerical solution of this task can be defined for example as a general non-linear optimization program as shown:

**FCEI-OPP:** Optimization program for estimation of ultimate causal capacity $C_U$ and intensity $i$ from observed data:

- Estimate $E(C_o)$, $dE/dC_o$, and $d^2E/dC_o^2$ (11-20) at the moment of observation (e.g. apt fitting to observed data (Fig. 2)).
- Estimate interaction potential $W_o$ (e.g. by numerical integration of observed data (Fig. 2)).
15  - Set objectives $W_F(C_o, C_U, i) = W_o$ and $d^2E/dC^2 = d^2E/dC_o^2$ by changing $C_U$ and $i$ in (17) as free variables.



## 4 Climate change and ice melting interaction

Decades of researches of climate dynamics and climatic change relating cryosphere have been highlighting many complex hardly jointly manageable interrelated causes and effects as well as the importance of a number of various feedbacks and interactions of the land ice sheets, the sea ice, the sea level elevation, atmosphere, hydrosphere, oceans and the global climate

system. The meanings of multiple feedback mechanisms and interactions giving rise to the aperiodic oscillations in climate systems including ice-albedo feedback, precipitation-temperature feedback and interactions between the ice sheets and the bedrock has been earlier identified in investigation of climatic change (Yiou et al., 1994). The study of the Greenland surface ice mass balance confirmed the significance of the feedback between the surface climate and the surface albedo in energy-balance-based ablation calculations of interest for climate dynamics (Lefebre et al., 2005). Different interactions and feedbacks

of long-term ice sheet-climate and anthropogenic climate change were studied in order to investigate the sea level rise and the impact of ice sheet changes on the climate system (Vizcaino et al., 2008). Ice-climate interactions and climate sensitivity were also investigated by considering the Greenland and Antarctica ice sheet climate dynamics (Goelzer et al., 2011). The relevance of atmosphere and ice sheet interaction is recognized for study of ice sheet stability and ice mass balance formulation (Solgaard and Langen, 2012). The interactions of land ice sheets and climate has a long history and it is investigated on long time scale

accounting for ice-albedo and surface elevation feedback also accounting for the influence of CO2 and insolation by transient simulation of the past 800000 years (Stap et al., 2014). Moreover, researchers take for granted that the interaction between the climate system and the large polar ice sheet regions is a key process in global environmental change regarding the cryosphere (Gong, Cornford and Payne, 2014).

The GRACE (Gravity Recovery and Climate Experiment) time-variable gravity satellite observations on a monthly basis from

April 2002 to June 2017 (Velicogna and Wahr, 2006a; Velicogna and Wahr, 2006b; Velicogna, 2009; Sasgen et al., 2013; Velicogna et al., 2014; Tedesco et al., 2015; Tedesco et al., 2016; Tedesco et al., 2017; Wiese et al., 2016) indicated supra-linear growth of annual average rates of ice mass losses in the period of $T_s=15^2/_{12}$ years of about 286 $Gt/year$ for the Greenland ice sheet and 127 $Gt/year$ for the Antarctica ice sheet, respectively (for example Figs. 3 and 4).

For all these reasons this study aims to investigate whether the general concept of finite causal processes formulated by the

FCEI mathematical model (11-20) in this article can confirm the relation of the climate change feedback $F$ and interaction $I$ between ice malting $M$ (the cause $C$) and ice mass losses $L(M)$ (the effect $E$) as finite dynamical causal process (FMLI).

### 4.1 Finite ice melting and ice mass losses mathematical model

The primary ice mass losses $L'[M(T)]$ in time $T$ are related of the primary ice melting $M(T)$ according to (1) as it is shown:

$$L'\big[M(T)\big] = M(T) = t \cdot T \tag{21}$$

The rate $t=L/T$ in $Gt/year$ of primary ice mass loss $L$ during observation time $T$ at the beginning of systematic data collection indicates the initial propensity to ice mass losses $L[M(T)]$ during ice melting $M(T)$ (Figs. 3 and 4) in T-L and M-M-L spaces.



The remaining ice mass $R(M)$ after an observed mass $M$ of ice has melted out of the total finite mass $M_U$ of ice sheets is defined following (11) as shown

$$R(M) = M_U - M = M_U \cdot (1-m) \tag{22}$$

In (22) $m = M/M_U = tT/M_U$ is the observed ice melting $M$ relative to the supposed ultimate mass $M_U$ of ice sheets.

### 4.2 Interaction of environmental thermal energy and ice sheet heat capacity

The reported acceleration of ice mass losses alarmed that an increasing amount of environmental heat $H[M(T)]$ of the climate system has been transferring to the melting mass $M$ of ice sheets during the observation time $T$. Simultaneously the ice sheets during this heat transfer have been losing their total inherent finite heat capacity $Q(M_U)$. The reduction of the heat capacity $Q[M(T)]$ (e.g. in $GJ$) of the ice sheets may be considered in proportion $h$ to the mass of melted ice $M$ in time $T$ as shown:

$$Q[M(T)] = h \cdot M(T) \tag{23}$$

In (23) $h$ is the specific heat capacity of ice at some temperature $K$ (e.g. 2.108 x $10^3$ ($GJ/(Gt\,K)$) or particularly the specific latent heat of fusion $h$ (e.g. 3.34 x $10^8$ $GJ/Gt$) during transition from solid state of ice to liquid state.

The diminishing residual heat capacity $Q[M_U-M(T)]$ after the mass $M$ of ice has melted in time $T$ is proportional to the remaining ice mass of the ice sheets $M_U-M$ (22) as shown:

$$Q[M_U - M(T)] = h \cdot [M_U - M(T)] \tag{24}$$

The character of the reported ice mass anomaly suggests that the heat flow from the environmental thermal energy plausibly intensifies at least linearly in proportion $i$ to the loss of the heat capacity $Q[M(T)]$ (23) of ice sheets as shown:

$$H[M(T)] = i \cdot h \cdot M(T) \tag{25}$$

The parameter $i$ in (25) represents the heat transfer interaction intensity between the environmental heat energy of the climate system and the heat capacity of ice sheets.

### 4.3 Finite interaction of ice melting and ice losses

The rate of the heat transfer from the environmental climate system to the ice sheets is defined according to (12) as the ratio of the increasing thermal energy $H[M(T)]$ (25) of the environmental climate system and the diminishing residual heat capacity $Q[M_U-M(T)]$ (24) of the ice sheets as follows:

$$\frac{dI_F[L,M(T)]}{dT} = \frac{H[M(T)]}{Q[M_U-M(T)]} = i \cdot \frac{m}{1-m} \cdot t \tag{26}$$

The last term (26) expresses the interaction between the climate system heating and the melting of ice sheets exactly in the same form of the FCEI interaction rate as in (12) but now expressed by relation of melted $M$ and remaining $M_U-M$ (22) ice masses. The term (26) expresses the physical rate of heat energy flow from the climate system to the ice sheets in term of the melted ice mass and thermal energy equivalencies (23-25) (Figs. 3 and 4).





Hence, the study applies in the sequel the FCEI mathematical model (11-20) with the aim to investigate whether the relations of climate change and ice mass losses of ice sheets may be considered as finite dynamical causal processes.

The integration of (26) yields to the FMLI interaction term as in (13) where the non-dimensional function $u(m)$ (14) represents the effect of the ultimate ice mass $M_U$ to ice melting $M$ in T-L space o in M-L space (Figs. 3 and 4) as follows:

$$I_F\left[L, M(T)\right] = \int_0^M i\, \frac{M(T)}{M_U - M(T)}\mathrm{d}M = i \cdot u(m) \cdot M = i \cdot u(m) \cdot t \cdot T \tag{27}$$

The overall ice mass loss $L[M(T)]$ yields to the FMLI relation following (15) as shown:

$$L\left[M(T)\right] = L'\left[M(T)\right] + I_F\left[L, M(T)\right] = \left[1 + i \cdot u(m)\right] \cdot t \cdot T \tag{28}$$

The theoretical finite interaction potential $W_F[I_F(L,M(T))]$ during ice melting $M$ exposed to climate conditions is expressed by the integral of (27) where $w(m)$ is given in (17), respectively, as shown:

$$W_F\left[I_F(L, M(T))\right] = \int_0^T I_F\left[L, M(T)\right]\mathrm{d}M = M_U \cdot i \cdot w(m) \cdot T \tag{29}$$

The overall theoretical finite potential during ice melting $M$ exposed to climate conditions implies the potential of primary ice melting (21) and interaction potential (29) as shown:

$$W\left[L, M(T)\right] = \int_0^T L\left[M(T)\right]\mathrm{d}M = M_U \cdot \left[m/2 + i \cdot w(m)\right] \cdot T \tag{30}$$

The theoretical total finite potential $W[L, M(T)]$ (30) represents the mass equivalence (e.g. in $Gt$ x $year$) to the overall accumulated real work done by all environmental effects, that is, the absorbed external thermal energy $H[M(T)]$ (25) of all sources of the climate system on expense of the loss of the diminishing heat capacity $Q[M(T)]$ (23) during the melting of mas $M$ of the ice sheets in time $T$. Negative intensity interaction parameter $i$ in (21-30) indicates possible recovery.

The first derivative of the ice mass losses $L[M(T)]$ (28) is the rate of change of the interaction (27) of the FMLI as follows:

$$\mathrm{d}L\left[M(T)\right]/\mathrm{d}T = \left(p + i \cdot \frac{m}{1-m}\right) == \left(p + i \cdot \frac{m}{1-m}\right) \cdot t \tag{31}$$

The second derivative of the ice mass losses $L[M(T)]$ (28) is the rate of change of the interaction (27), that is, the sensitivity of the FMLI as follows:

$$\mathrm{d}^2 L\left[M(T)\right]/\mathrm{d}T^2 = \mathrm{d}I_F\left[L, M(T)\right]/\mathrm{d}T = \frac{i}{M_U} \cdot \frac{1}{(1-m)^2} \cdot t^2 \tag{32}$$

The time of the beginning of the intense ice mass anomaly $T_B$ before the start of observations at $T_o$ follows from the condition of the minimal rate of ice mass losses $\mathrm{d}L/\mathrm{d}T=0$ in (28) and can be calculated as shown:

$$T_B = T_o - \frac{M_U}{t} \cdot \frac{1}{(i-1)} \tag{33}$$

The total melting out time $T_M$ for the ultimate loss of the ice mass $M_U$ of ice sheets is the numerical solution of the inverse of the equation (28) $T_M = L^{-1}(M_U)$. The fracture of interaction potential with respect to total potential is simply $W_F/W$.




The total potential to finite melting out for $m_M=(t\,T_M)/M_U$ in (30) and for full melting out time $T=T_M$ amounts to $W_M=M_U$ $[m_M/2+i\,w(m_M)]\,T_M$. The ratio of $W_F$ (29) and $W$ (30) indicates the relative importance of the interaction potential.

The total mass $M_U$ of ice sheets and the interaction intensity $i$ are determined by the optimization program FCEI-OPP by the following numerically procedure:

• Estimate $M_o$, $\mathrm{d}L/\mathrm{d}M_o$, $\mathrm{d}^2L/\mathrm{d}M^2$ and $t$ at the start of observation $T_o$ (e.g. appropriate fitting to observed data (Fig. 2)).

   • Estimate the total potential $W_o$ of the process (30) (e.g. by numerical integration of observed data (Fig. 2)).

   • Apply FCEI-OPP: Fulfil the above equality $W\big[L,M_o,M_U\big]=Wo$ by changing $M_U$ as a free variable.

   • The interaction intensity is then $i=M_U\,\mathrm{d}^2L/\mathrm{d}M^2$.

The FMLI mathematical model (21-34) is applied in the sequel to investigate the interactions between the ice mass loss of the

Greenland and Antarctica ice sheets and the climate change (Figs. 3 and 4). The following analysis gratefully uses the data from the GRACE JPL RL05M.1 Mascon Solution: Version 2 provided by Wiese et al. (2016).

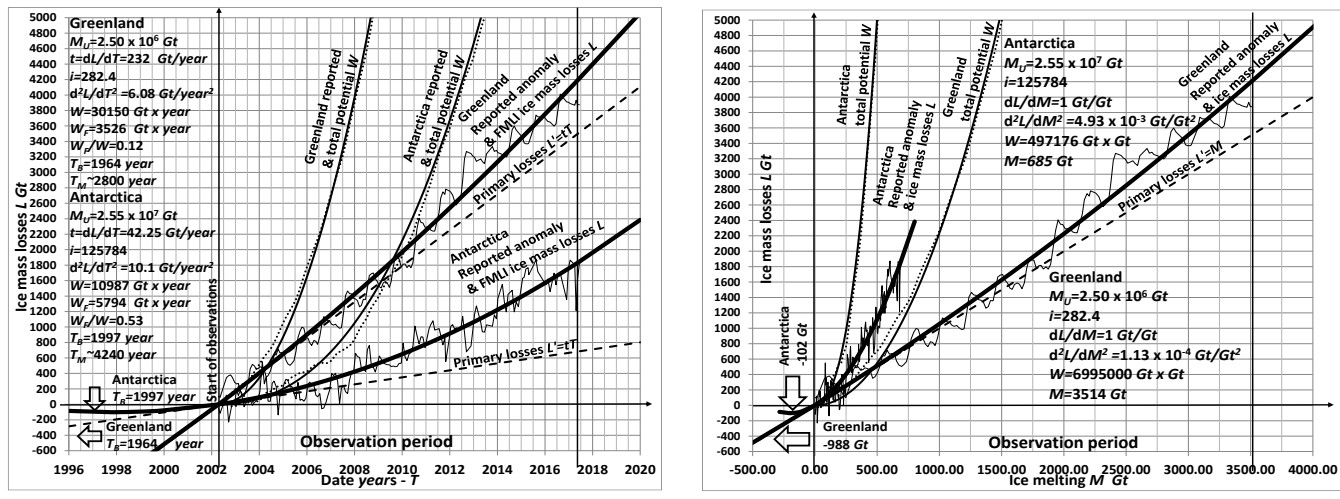

**Figure 3: Greenland and Antarctica reported ice mass anomaly analysis and FMLI numerical results in T-L and M-L spaces.**

**4.4 Interaction of ice melting and ice mass losses of the Greenland ice sheet.**

Volume and mass of ice sheets are not precisely known. The density of glacier ice is estimated in the range of $\gamma_{ice}$=0.900-0.917 $Gt/km^3$. Greenland land ice sheet volume is estimated from different sources at $V_U$=2.50-3.00 x $10^6$ $km^3$ and the total ice mass in the range $M_U$= 2.5±0.20 x $10^6$ $Gt$ (±8%). The Greenland sheet ice mass using a coherent ice-penetrating radar system to produce a thickness grid is estimated to 2.93 x $10^6$ $km^3$ or to about 2.65 x $10^6$ $Gt$ (Bamber et al., (2001).

A.   The analysis of observed data by Wiese et al. (2016) provides the total external work done by all environmental effects, in terms of mass equivalence to the thermal energy of the climate system absorbed during $15^2/_{12}$ $years$ of melting of the ice sheet that equals to the potential (30). It is estimated by numerical integration of observed data using the trapezium rule in amount of $W_o[T=15^2/_{12}\ years)$]=30150.7 $Gt$ x $year$ in T-L space (30) or 6995000 $Gt$ x $Gt$ in M-L space (Fig. 3).



B.  Preliminary fitting of the 2nd order parabola in T-L space $L[M(T)]=aT^2+bT+c$ (Fig. 5) to the observed data that satisfies the above condition for total observed potential (30) $W_o[T=15^2/_{12} \text{ years})]= 30150.7$ *Gt* x *year* provides $a$=3.04, $b$=232 and $c$=0.09. Fitting of the 2nd order parabola cannot provide information on ultimate mass $M_U$ of the ice sheet.

C.  For calculated total potential (30) $W[T=15^2/_{12} \text{ years})]$=30150.7 *Gt* x *year* same as the observed potential $W_o$ and for estimated acceleration $d^2L/dT^2=2a$=6.08 (*Gt/year*)*/year* (32), the optimization program FCEI-OPP provides the FMLI interaction intensity parameter $i$=282.4 and the total ice mass of Greenland ice sheet of $M_U$= 2.50 x $10^6$ *Gt* (Fig. 3). This is a good estimate within the assumed range $M_U$= 2.5±0.20 x $10^6$ (±8%) and with respect to 2.65 x $10^6$ *Gt* (Bamber et al. (2001). The coefficient of determination *R-squared* between observed data and FMLI curve is $R^2$=0.978.

D.  Final analysis by using the FMLI model provides additional information on Greenland ice mass melting.

•  The rates of ice mass loss $L$ (31) in Greenland increased from $dL/dT$~230 *Gt/year* in 2002 to ~275 *Gt/year* in 2009 and to ~325 *Gt/year* at the end of observation in 2017 confirming the nonlinear character of the ice mass loss (Fig. 3) at an average of ~4133/$15^2/_{12}$ =280 *Gt/year* or about 23 *Gt/month* during the observation period. Reported average is 286 *Gt/year*, Wiese et al. 2016).

•  The alarming acceleration of ice mass loss rates (32) starts at the beginning of the observations in the amount of
$d^2L/dT^2$=6.08 (*Gt/year*)*/year* and slightly increasing in the future. The total observed ice mass loss is 4212 *Gt*.

•  The interaction part (29) of the total potential (30) of the climate system and the ice melting is calculated in amount of $W_F[T=15^2/_{12} \text{ years})]$=3526 *Gt* x *year* (29), that is about 12% of the total potential (above the primary losses line Fig. 3).

•  By extrapolating the FMLI curve (28) to the past time, the estimated beginning of the intensive ice mass anomaly on Greenland relative to the start of observations at the date $T_B$=2002$^4/_{12}$ – 38$^4/_{12}$~1964 (33), (Fig. 3).

•  Since the beginning of intensive losses to the start of observation on Greenland during 38$^4/_{12}$ years about 988 *Gt* of ice have been already lost. To the end of observation during 38$^4/_{12}$+15$^2/_{12}$=53$^{10}/_{12}$ years about 988+4212=5200 *Gt* of ice is lost, what is 0.2% of the total ice mass $M_U$ (Fig. 3).

•  It is possible to predict by extrapolating the FMLI ice mass loss curve (28) that the melting out of the total mass of ice $M_U$= 2.50 x $10^6$ *Gt* due to the interaction with climate change under same environmental conditions could happen in the
year $T_M$=2850±70 with 8% uncertainty of ultimate ice mass $M_U$ estimation (Figs. 3 and 4).

## 4.5 Interaction of ice melting and ice mass losses of the Antarctica ice sheet

The Antarctica ice sheet volume is estimated at 2.5-3.0 x $10^7$ *km³*. The total ice mass is assumed in the range $M_T$=2.5±.25 x $10^7$ *Gt* (±10%). The Antarctica sheet ice mass is estimated by building digital topographic models from long time collection of ice thickness data to 25.4 *km³* what is about 2.3 x $10^7$ *Gt* (Lythe et al., 2001).

A.  The analysis of observed data by Wiese et al. (2016) provides the total external work done during $15^2/_{12}$ *years* of melting of the Antarctica ice sheet by numerical integration of observed data by using the trapezium rule in an amount of $W_o$=10987.3 *Gt* x *year* in T-L space (30) or 497176.6 *Gt* x *Gt* in M-L space (Fig. 3).



B.  Preliminary fitting of the 2nd order parabola (Fig. 3) for potential $W_o$ (30) provides $a$=5.05, $b$=45.25 and $c$=-3.9.

C.  For calculated total potential (29) $W_o[T=15^2/_{12} \ years)]$=10987.3 $Gt$ x $year$ equal to the observed $W_o$ and for estimated acceleration $d^2L/dT^2$=2$a$=10.1 $(Gt/year)/year$, the optimization program FCEI-OPP provides the FMLI interaction intensity parameter $i$=125784 and the total ice mass of Antarctica ice sheet of $M_U$= 2.55 x 10$^7$ $Gt$ (Fig. 3). This is a good
estimation within the reported range $M_T$=2.5±0.20 x 10$^7$ $Gt$ (±8%) and with respect to other sources, e.g. 2.65 x 10$^7$ $Gt$ Lythe et al. (2001). The coefficient of determination $R$-$squared$ between observed data and FMLI curve is $R^2$=0.978.

D.  Final analysis by using the FMLI model provides additional information on Greenland ice mass melting.

•  The rates of ice mass loss $L$ (31) in Antarctica increased from d$L$/d$T$~50 $Gt/year$ in 2002 to ~120 $Gt/year$ in 2009 and to ~198 $Gt/year$ at the end of observation in 2017 (Fig. 3). This is an average rate of ~1844/15$^2$/$_{12}$ =122 $Gt/year$ or about 10
$Gt$/$month$ in the observation period. Reported average is 127 $Gt/year$ (Wiese et al., 2016).

•  The alarming acceleration of ice mass loss rates (32) starts at the beginning of the observations in the amount of d$^2L$/d$T^2$=10.1 $(Gt/year)/year$ and slightly increases in the future.

•  The interaction part (29) of the total potential (30) of the climate system and the ice melting process is $W_F[T=15^2/_{12}$ $years$)]=5794 $Gt$ x $year$ (29), that is about 53% of the total potential (above the line of preliminary losses at Fig. 3).

•  By extrapolating the FMLI curve (28) to the past time, the estimated beginning of the intensive ice mass anomaly in Antarctica relative to the start of observations at the date $T_B$=2002$^4$/$_{12}$ − 4$^6$/$_{12}$ ~1997$^{10}$/$_{12}$ (33), (Fig. 3). Since the beginning of intensive losses to the start of observation in Antarctica during 4$^6$/$_{12}$ years about 102 $Gt$ of ice have been already lost. To the end of observation during 4$^6$/$_{12}$+15$^2$/$_{12}$=19$^7$/$_{12}$ years about 102+1845=1947 $Gt$ of ice is lost, what is 0.08% of the total ice mass $M_U$ (Fig. 3).

•  The melting out date of the total mass of ice $M_U$= 2.55 x 10$^7$ $Gt$ due to the interaction with climatic change could happen in the year $T_M$=4240±220 with 10% uncertainties of ultimate ice mass $M_U$ estimation (Figs. 3 and 4).

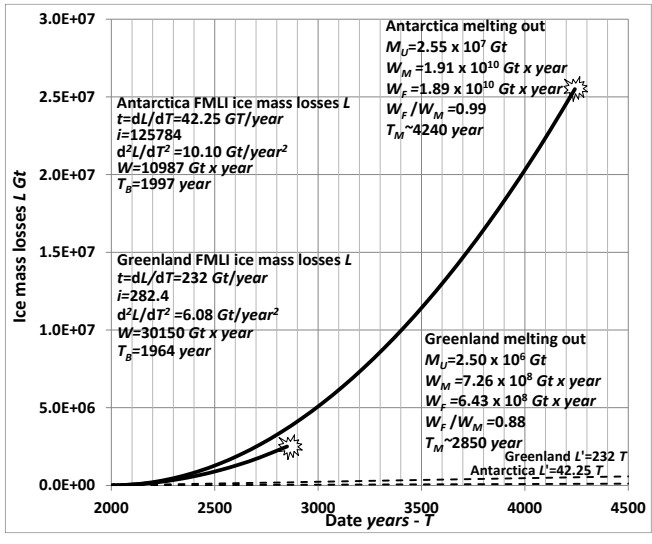
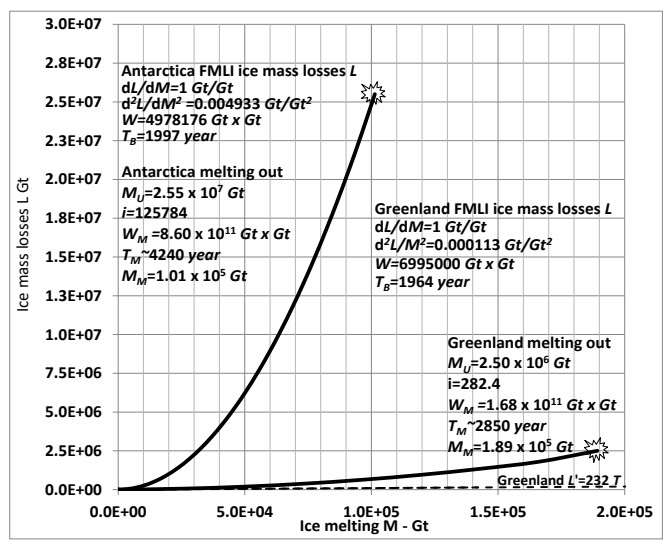

**Figure 4:** Greenland and Antarctica ice mass losses predictions of based on FMLI in T-L and M-L spaces.





## 5 Discussion

The analyses of total potentials $W$ (30) in the 17 years of observations indicate that short-termly all effects of combined external factors together with the intrinsic properties of Greenland ice sheet in amount of 30416 $Gt$ x $years$ exceeds 2.75 times the total potential of Antarctica ice sheet in amount of 10987 $Gt$ x $years$ (Fig. 3). The overall long term predictions until total melting

out of entire ice sheets indicate that the overall potential of $W_M$=1.90 x $10^{10}$ $Gt$ x $years$ of Antarctica during 2240 years about 26 times exceeds the overall interaction potential of Greenland in amount of $W_M$=7.24 x $10^8$ $Gt$ x $years$ during 850 years. The importance of interaction potentials $W_F$ (29) increase from observed 12% and 53% to 88% and 99% in total potential $W_M$ (30) towards predicted melting out of Greenland and Antarctica ice sheets, respectively (Fig. 4).

The two curves of ice mass losses, 2nd order parabola and the FMLI curve (28) almost coincide in the range of observed data

with high coefficient of determination $R$-$squared$: $R^2$=0.999 but slightly diverge in the future due to effect of finiteness. Since the observed mass of ice melting represent only a very small part ($10^{-3}$-$10^{-4}$) of the total ice mass, the precision of the results of the inverse numerical calculations with the FMLI mathematical model (21) and (22) for estimation of the total ice sheets masses of Greenland $M_U$=2.54 x $10^6$ $Gt$ and of Antarctica $M_U$=2.50 x $10^7$ $Gt$, need to be considered cautiously.

## 6 Conclusion

Finiteness is the fate of the world we know by experience. Therefore this study investigated the trans-temporal finite interactions of effects and causes simultaneously affecting and being affected by the limited and exhaustible causal capacities typical for finite causal processes. The mathematical model of finite causal processes developed during this studies relate both the observable past and the imaginable future separated by the present moment of observation. The link between the elapsed effects and forthcoming causes relies on the experience that wasting of limited causal capacities in the past continues with

conceivable regularity of observed conditions until running out of the finite resources in future.

The finite cause-and-effect interaction concept elaborated in this study recognises the climate dynamics of the recently observed ice mass anomaly as a finite trans-temporal causal continuum between intensified climatic change and accelerated ice mass losses of a limited amount of irrecoverably diminishing residual mass of Greenland and Antarctica ice sheets in complex but restricted natural conditions of the cryosphere.

The theoretically founded interaction potential depends on known ultimate mass of Greenland and Antarctica ice sheets and on interaction intensity that can be estimated from observed data during melting of a limited mass of ice. However, the assumption of the finiteness and the inverse application of the mathematical procedure of finite interaction model makes it possible to re-estimate the total mass of ice sheets from the observed ice mass anomaly data.

The study holds that the interaction concept of finite dynamical causal processes in the article well describes the intricate

relation between the observable cryosphere ice mass anomaly of the Greenland and Antarctica ice sheets and the acceleration of climate change under combination of hardly jointly manageable interactions and feedbacks of different aggregate intrinsic, environmental, natural and human induced circumstances of finite and vulnerable planetary resources.



**Nomenclature**

| | | |
|---|---|---|
| $C$ – cause, in general | $M$ – ice melting | c – cause, relative |
| $C_U$ – cause, ultimate | $M_U$ – ice mass, ultimate | f – feedback factor |
| $E$ – effect, in general | $Q$ – heat capacity of ice | i – interaction intensity |
| $F$ – feedback, in general | $R$ – residual cause | m – ice melting, relative |
| $H$ – environmental thermal energy | $T$ – time, in general | h – specific heat capacity of ice |
| $I$ – interaction, in general | $T_o$ - time, start of observation | p – rate of change, open-loop gain |
| $I_F$ – interaction, finite | $T_B$ – time, beginning of ice losses | q – closed-loop gain |
| $K$ – temperature | $T_M$ – time, melting out | t – rate of ice mass loss in time |
| $L$ – ice mass loss | $W$ – potential, in general | u – finite interaction correction function, |
| | $W_F$ – interaction potential, finite | w – finite interaction potential correction function |

**Appendix A. Derivatives of functions with respect to bounds of finite variables**

Consider a function $f(x)$ where the variable $x$ is limited on positive side by its upper bound $X$.

The infinitesimal change of the function $f(x)$ with respect to the finite complementary variable $X$-$x$ relative to its upper bound

5   $X$ is geometrically interpreted (Fig. A-1) as follows:

$$\frac{\Delta f(x)}{\Delta(X-x)} = \frac{f(x) + \Delta f(x)}{X - x} \tag{A-1}$$

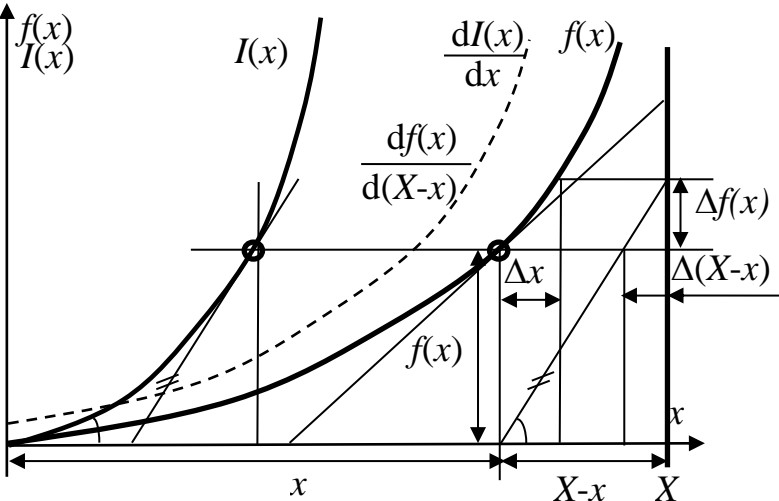

**Figure A-1: Infinitesimal change of function with respect to upper bound of finite variable.**

10   The derivative of the function $f(x)$ with respect to the finite complementary variable (X-x) relative to its upper bound is

reinterpreted by limit of the slope of the function $I(x)$ with respect to the variable $x$ as follows:

$$\frac{dI(x)}{dx} = \frac{df(x)}{d(X-x)} = \lim_{\Delta(X-x)\to 0} \frac{\Delta f(x)}{\Delta(X-x)} = \lim_{\Delta(X-x)\to 0} \frac{f(x) + \Delta f(x)}{X - x} = \frac{f(x)}{X - x} \tag{A-2}$$



The function $I(x)$ represents the effect of finiteness and itself then can be obtained by integration of (A.2) as shown:

$$I(x) = \int_0^x \frac{df(x)}{d(X-x)} dx = \int_0^x \frac{f(x)}{X-x} dx \tag{A-3}$$

The second derivative is then:

$$\frac{d^2 I(x)}{dx^2} = \frac{d^2 f(x)}{d(X-x)^2} = \frac{1}{(X-x)^2} \left[ \frac{df(x)}{dx} \cdot (X-x) + f(x) \right] \tag{A-4}$$

The area below I(x) is the integral of (A-3) as follows:

$$W(x) = \int_0^C I(x) dx \tag{A-5}$$

Note that for given values of $I(x)$, $W(x)$, $dI(x)/dx$ and $d^2I(x)/d^2x$ is possible to find $X$ by solving equations A-1 to A-5.

Example 1. The first example is the linear or 1$^{st}$ order causal relation $E(C)=C$ used by the FCEI model in the text (11-20).

Example 2. The second example is the quadratic or 2$^{nd}$ order primary causal relation (1) (Fig. A-2) defined as shown:

$$E'(C) = C^2 \tag{A-6}$$

According to (12) and (A-2) the rate of change of the interaction between a cause $C$ and an effect $E$ is:

$$\frac{dI(E,C)}{dC} = \frac{C^2}{C_U - C} \tag{A-7}$$

The interaction of finite causal relation itself according (13) represents the effect of finiteness is following (A-3) is the integral of (A-7) as show:

$$I(E,C) = \int_0^C \frac{C^2}{C_U - C} dC = -C \cdot (C_U + C/2) + C_U^2 \ln \frac{C_U}{C_U - C} \tag{A-8}$$

The second derivative of interaction according (20) and (A-4) is then:

$$\frac{d^2 I(E,C)}{dC^2} = \frac{d^2 E(C)}{d(C_U - C)^2} = C \cdot \frac{2 \cdot C_U - C}{(C_U - C)^2} \tag{A-9}$$

The interaction potential is the integral of according (17) and (A-5) is then:

$$W[I(E,C)] = \int_0^C I(E,C) dC = -C_U \cdot \left[ I(E,C) + C^2 \right] + C_U^2 \cdot C \cdot \ln \frac{C_U}{C_U - C} - C^3/6 \tag{A-10}$$

For example, for given $C=0.75$ and $I(E,C)=1$ (A-8) the calculated finite causal capacity is $C_U=0.7889$ (Fig. A-2).





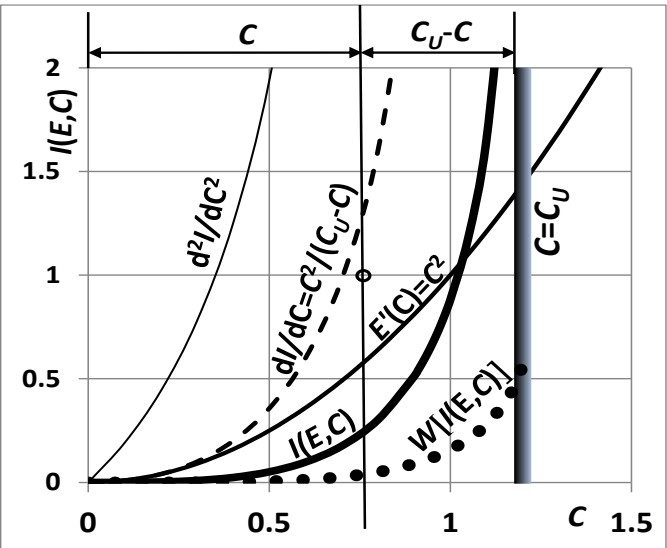

**Figure A-2: Example of a second order finite causal relation $E(C)=C^2$ for $C_U=1.2$.**

5   *Acknowledgements*

*Dataset:* "The Group for High Resolution Sea Surface Temperature (GHRSST) Multi-scale Ultra-high Resolution (MUR) SST data were obtained from the NASA EOSDIS Physical Oceanography Distributed Active Archive Center (PO.DAAC) at the Jet Propulsion Laboratory, Pasadena, CA (http://dx.doi.org/10.5067/GHGMR-4FJ01)."

*Tools and Services:* "The Group for High Resolution Sea Surface Temperature (GHRSST) Multi-scale Ultra-high Resolution

10   (MUR) SST data were obtained from the Live Access Server (LAS) at the NASA EOSDIS Physical Oceanography Distributed Active Archive Center (PO.DAAC), Jet Propulsion Laboratory, Pasadena, CA (https://podaac.jpl.nasa.gov/las).



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
