# Peer review of "Dynamics of finite causal processes"

_Earth System Dynamics, 2018_

## Referee Comment (RC1) · Anonymous Referee #1 · 15 May 2018

REJECT

This manuscript attempts to apply the concepts of the 'general systems theory' (GST) of Ludwig von Bertalanffy to explain changes of the physical climate physical, in particular the causal relationship between 'finite ice melting' (the cause M) and 'ice mass Losses' (The effect). The 'general systems theory' of Ludwig von Bertalanffy is a holistic, controversial theory, started in 1938, that may qualitatively explain some phenomena in ecology and social sciences but is far from being accepted as a science subjected to verification and falsification. Moreover, it presents a simplistic theory of the linear feedback which comes out as a very particular case of the much more general and well-grounded and mathematically funded 'Control system's theory'. The application of GST formalism to the relationship between 'Finite ice Melting' and 'ice mass Losses' seems therefore inappropriate giving rise to 'vague' concepts without any physical correspondence. Examples of that are quoted from the manuscript:

[Figure]

1) 'This study of finiteness of natural processes recognizes the temporal FCEI (Finite Cause-and-Effect Interaction) empirical concept as a continuous sharing of irreplaceable and restricted overall ultimate causal capacity CU between the observable elapsed effect E in the past and the imperceptible but conceivable forthcoming limited exhaustible cause C beyond the instant of observation in the future.

2) The trans-temporal finite interaction implies the empirical link in continuation of the known uninterrupted past and the imaginable but finite perpetuating future separated by the instant of observation at the present time'

3) The mathematical model of the FCEI in this study considers a simple intuitive term of the continuous residual causal capacity R(C) after spending some primary effect E'(C) (1) of the limited cause C on the expense of the ultimate cause CU.

Beyond the above criticisms, the author tries to make millenary climatic predictions (extrapolations) using the simplistic GST relationships, as quoted from the manuscript:

'It is possible to predict by extrapolating the FMLI ice mass loss curve (28) that the melting out of the total mass of ice MU= 2.50 x 106 Gt due to the interaction with climate change under same environmental conditions could happen in the year TM=2850ïĆś70 with 8% uncertainty of ultimate ice mass MU estimation (Figs. 3 and 4)'.

That prediction is totally speculative and cannot be accepted.

Moreover, the author does totally ignore alternative approaches studying the causality in the climatic system (e.g. Granger causality) and therefore it is not understood in which the manuscript adds new knowledge.

Giving the above arguments, the manuscript must be rejected and cannot be accepted to 'Earth System Dynamics' journal.
* * *

---

## Author Comment (AC1) · 21 May 2018

These are the comments on discussion submitted by anonymous Referee #1 for manuscript entitled 'Dynamics of finite causal processes'

Referee: This manuscript attempts to apply the concepts of the 'general systems theory' (GST) of Ludwig von Bertalanffy to explain changes of the physical climate, in particular the causal relationship between 'finite ice melting' (the cause M) and 'ice mass Losses' (The effect). The 'general systems theory' of Ludwig von Bertalanffy is a holistic, controversial theory, started in 1938, that may qualitatively explain some phenomena in ecology and social sciences but is far from being accepted as a science subjected to verification and falsification. Moreover, it presents a simplistic theory of the linear feedback which comes out as a very particular case of the much more general and well-grounded and mathematically funded 'Control system's theory'.

[Figure]

Comment: The manuscript evokes the formalism of the 'general systems theory' (GST) of Ludwig von Bertalanffy (1938) in order to put the concept of finite causality in systemic framework. It also recalls the mathematical structure of the control system theory (CST) of Wiener (1948) (Eqns. 1-5 and Fig. 1) as a platform for decomposition of general causal relations to primary proportional and to appropriate interaction parts (Eqns. 6-10 and Fig. 1). This decomposition is needed for definition of continuous Finite Cause-and-Effect-Interaction (FCEI) concept that is mathematically elaborated in the manuscript (Eqns. 11-20 aind Fig. 2). Hence, the article extends the concept of simplistic unlimited linear feedback of the CST by the novel FCEI concept of additional feedback induced solely by the fact of finiteness of processes with exhaustible causal capacities.

Referee: The application of GST formalism to the relationship between 'Finite ice Melting' and 'ice mass Losses' seems therefore inappropriate giving rise to 'vague' concepts without any physical correspondence. Examples of that are quoted from the manuscript.

Comment: Instead of the GST formalism, the combined FCEI approach (Eqns. 11-20 and Fig. 2) is applied in the manuscript to investigate the relationships between 'Finite ice Melting' and 'Ice mass Losses' in Interaction with climate system (FMLI) (Eqns 21-33 and Figs. 3-4). The FMLI concept in the article accounts for physical interchange of environmental heat energy between climate system and inherent finite heat capacity of ice sheets (Eqns. 23-26). The examples quoted by the reviewer as 'vague', in the manuscript represent the wordily interpretations either of mathematical terms (Eqns. 11-20 and 21-33) or of figures (Figs. 1-2). Here there are.

Referee: 1) 'This study of finiteness of natural processes recognizes the temporal FCEI (Finite Cause-and-Effect Interaction) empirical concept as a continuous sharing of irreplaceable and restricted overall ultimate causal capacity CU between the observable elapsed effect E in the past and the imperceptible but conceivable forthcoming limited exhaustible cause C beyond the instant of observation in the future. 2) The
trans-temporal finite interaction implies the empirical link in continuation of the known uninterrupted past and the imaginable but finite perpetuating future separated by the instant of observation at the present time'

Comment: This two statements in the manuscript wordily explains the temporal character of the mathematical rate of change E(C)/(CU-C) in the FCEI definition (Eqn. 12) as the ratio between the growing elapsed effect E(C) in the past and the diminishing remaining driving cause (CU-C) with respect to the waste ultimate causal capacity CU in the future separated by the instant of observation (Fig. 2). 3) The mathematical model of the FCEI in this study considers a simple intuitive term of the continuous residual causal capacity R(C) after spending some primary effect E'(C) (1) of the limited cause C on the expense of the ultimate cause CU.

Comment: This statement represents the conservation principle between elapsed effect and forthcoming cause which expresses the assumption that a continuous finite causal relation does not change its properties of propensity to and intensity of interaction during unchanged environmental conditions with respect to wasting of ultimate causal capacity CU over time (Eqns. 11-20).

Referee: Beyond the above criticisms, the author tries to make millenary climatic predictions (extrapolations) using the simplistic GST relationships, as quoted from the manuscript: 'It is possible to predict by extrapolating the FMLI ice mass loss curve (28) that the melting out of the total mass of ce MU=2.50x106 Gt due to the interaction with climate change under same environmental conditions could happen in the year TM=285070 with 8% uncertainty of ultimate ice mass MU estimation (Figs. 3 and 4). That prediction is totally speculative and cannot be accepted.

Comment: The mathematical model of finite causal processes provides analytic parameters based on observed data which allow calculations beyond the instant of observation. Some of these results are exemplarily presented in the manuscript (Fig. 4). Following the reviewer's comment this extrapolations should be more appropriately

denoted as calculations than predictions.

Referee: Moreover, the author does totally ignore alternative approaches studying the causality in the climatic system (e.g. Granger causality) ...

Comment: Granger statistical causality investigates causal dependency between stochastic variables and do not indicate real causality in the deterministic context of the manuscript. Instead of Granger statistical concept of causality, the manuscript focuses on deterministic physical interpretations of work done by environmental effects and inherent properties of ice sheets giving the propensity, sensitivity and interaction intensity parameters of the mathematical model by derivatives and integration of the FCEI interaction curves (Eqns. 11-20 and Fig. 2). With respect to the scope and size of the manuscript no statistical analysis is planned.

Referee: ... and therefore it is not understood in which the manuscript adds new knowledge.

Comment: Submission of this manuscript is encouraged with declared aims and scopes of ESD journal. The manuscript presents a novel interdisciplinary approach to system dynamics that conceptualized, modelled and quantified the influence of finiteness of deterministic continuous causal processes with exhaustible capacities what was not applied earlier in investigations of Earth mechanisms. The manuscript introduces the finiteness as a property inseparable of other physical properties of continuous processes with limited causal capacities. It provides apt mathematical model of finite causal processes. The calculus in the manuscript implies a new general definition of derivatives of functions with respect to bounds of finite variables. The study also uncovers a new application of the mathematical model that makes it possible to discover and to estimate the unknown ultimate causal capacities from relevant information of supposedly finite causal processes beyond the instant of observation. The proposed analytics confirms that the climate system and ice mass anomaly monitored recently on Greenland and Antarctica ice sheets under global interaction of a combination of various component systems, such as the atmosphere, cryosphere, hydrosphere, oceans and human activities may be viewed as finite dynamical causal processes by definition given in the manuscript. Extrapolations suggest dates of beginning of intensive ice mass losses. The reverse numerical calculation procedure satisfactorily re-estimated the total ice mass of ice sheets within the suggested limits.

Referee: Giving the above arguments, the manuscript must be rejected and cannot be accepted to 'Earth System Dynamics' journal.

Comment: This discussion gives useful general comments that suggest additional clarifications which could hopefully improve the manuscript. What is missing is the analysis of the merit of the mathematical model and of the numerical analysis. A more detailed consideration of the concept and mathematical model of finite causal processes perhaps could give another view on intentions of the manuscript and on presented results. The introduction of finiteness in studies of natural phenomena is plausibly an important encouragement for understanding and analysis of finite processes in earth system dynamics. There are many potentially finite causal processes all over the world which might be studied by using coherent mathematical apparatus capable to deal with finiteness. Therefore, the publication of this submission with additional clarifications might be nevertheless useful for readers and researchers having interests in ESD journal.

---

## Author Comment (AC2) · 28 May 2018

**Appendix B Discrete interaction example of finite ice melting and ice mass losses**

This appendix brings forward an illustrative example calculation in 10 steps of the FMLI model (21-33) according the FCEI concept (11-20) in dimensionless form ($M_U$=1, $i$=1) presented in M-L space, (Table B-1 and Fig. B-1).

5  **Table B-1. Discrete numerical interaction model of ice melting in 10 steps**

| $M$ | $L'(M)$ | $M_U$-$M$ | $M/(M_U$-$M)$ | $I_F(L,M)$ | $L(M)$ | $W_F(M)$ | $M$ | $L(M)$ | $I_F/M$ | $W_F/W$ |
|-----|---------|-----------|---------------|------------|--------|----------|-----|--------|---------|---------|
|     | (21)*   | (22)      | (26)          | (27)       | (28)   | (29)     |     |        |         |         |
| 0.0 | 0.0 | 1.0 | 0.00 | 0.00 | 0.00 | 0.000 | ---------- |  | 0 | 0 |
| 0.1 | 0.1 | 0.9 | 0.11 | 0.05 | 0.15 | 0.000 | I--------- | Iₗ | 0.54 | 0.04 |
| 0.2 | 0.2 | 0.8 | 0.25 | 0.12 | 0.32 | 0.001 | II-------- | IIIₗ | 0.58 | 0.07 |
| 0.3 | 0.3 | 0.7 | 0.43 | 0.19 | 0.49 | 0.005 | III------- | IIIII | 0.63 | 0.12 |
| 0.4 | 0.4 | 0.6 | 0.67 | 0.28 | 0.68 | 0.014 | IIII------ | IIIIIII | 0.69 | 0.17 |
| 0.5 | 0.5 | 0.5 | 1.00 | 0.39 | 0.89 | 0.028 | IIIII----- | IIIIIIIII | 0.77 | 0.23 |
| 0.6 | 0.6 | 0.4 | 1.50 | 0.53 | 1.13 | 0.053 | IIIIII---- | IIIIIIIIII Iₗ | 0.88 | 0.30 |
| 0.7 | 0.7 | 0.3 | 2.33 | 0.72 | 1.42 | 0.094 | IIIIIII--- | IIIIIIIIII IIII | 1.03 | 0.38 |
| 0.8 | 0.8 | 0.2 | 4,00 | 1.01 | 1.81 | 0.158 | IIIIIIII-- | IIIIIIIIII IIIIIIII | 1.26 | 0.49 |
| 0.9 | 0.9 | 0.1 | 9,00 | 1.56 | 2.46 | 0.265 | IIIIIIIII- | IIIIIIIIII IIIIIIIIII IIIIₗ | 1.73 | 0.65 |
| 1.0 | 1.0 | 0.0 | ∞ | ∞ | ∞ | ∞ | IIIIIIIIII | IIIIIIIIII IIIIIIIIII IIIIIIIIIIIIIIIIII | ∞ | ∞ |

*Note: numbers in parenthesis () denotes the numbers of appropriate equations in the body text

The dimensionless term (26) $M/(M_U$-$M)$ (Table B-1) expresses the physical rate $H(M)/Q(M_U$-$M)$ of increasing environmental heat energy $H(M)$ (25) of overall climate system changes and the simultaneous losses of the thermal capacity $Q(M_U$-$M)$ (24)

10  of ice sheets due to melting of mass $M$ of ice. The positive heat flow accelerates the accumulation of ice mass losses $I_F(L,M)$ (27, integral of 26) due to interactions of ice melting $M$ and ice mass losses $L$ out of finite ultimate mass $M_U$. This acceleration is quantified by the interaction intensity parameter $i$ calculated from the observed data. The possible melting out point $M_M$ is presented in M-L space (Fig. B-1).

The overall ice mass losses $L(M)=L'(M)+I_F(L,M)$ (28) consist of primary losses (21) and losses due to interactions (27). The

15  relation $I_F/M$ expresses how much of ice mass $I_F(L,M)$ is lost due to interaction with respect to primary losses $L'(M)=M$. The interaction potential $W_F(L,M)$ (29, integral of 27) represents the amount of work done by all environmental feedbacks and interactions of climate system and ice sheets. The relation $W_F/W$ expresses how much work $W_F$ is done due to interactions with respect to work done on melting of primary losses, (Table B-1, Fig. B-1).

[Figure]

**Figure B-1. Model of discrete ice melting in 10 steps**

---

## Referee Comment (RC2) · Anonymous Referee #2 · 17 Jul 2018

Even though I am going to write a critical brief report, I strongly sympathize with the research questions that motivated this study, because they are extremely relevant and timely in general physics and in earth system dynamics in particular. The author takes a bold approach to causality that, at first sight, intuitively makes a lot of sense in view of open philosophical discussions that have been going on for a long time. However, those efforts are largely unfounded by physical science.

In my opinion, the fundamental problems are that the theoretical bases are far from being consolidated and the mathematical formulation is far from being fully matured. Moreover, the earth system dynamic applications appear to be speculative and incoherent with physical understanding of the earth system, as had been pointed out also by the other reviewer.

While I do recognize the philosophical and conjectural merits of the study and com-

mend the author for the strong investment in addressing such a tricky fundamental problem, I cannot recommend this manuscript for publication due to fundamental mathematical and physical concerns (which the other reviewer detailed carefully), especially the unproven validity of the theories and unproven physical reasonability of the formulations.

The author's second commend AC2 shows a significant effort to improve the manuscript and further clarify his points. However, to my understanding, while providing some practical added value, it does not overcome the major theoretical concerns of this manuscript.

Overall, when reading the paper I personally liked to read it from an informal conjectural perspective, though scientifically it was deemed unacceptable. Not because I disagree with the conjectures, but because the scientific method in all its thorough due process still needs to be conducted.

In my opinion, this manuscript could be submitted to a journal outside of the physical sciences, i.e. one where the intellectual exercise and theoretical conjectures themselves would be enough to grant publication. Or significantly matured in its mathematical and physical bases.

While conceptual research is always fascinating, here it should definitely be accompanied by a comprehensive in-depth proof of concept and validation at both mathematical and physical levels. A process that would require a mass of work beyond the scope of a major revision.

For this reason, I regret to inform that I am not recommending the manuscript for publication at Earth System Dynamics (ESD).

I appreciate your understanding and send my respects.
* * *

---

## Author Comment (AC3) · 7 Aug 2018

Reviewer #1: This manuscript attempts to apply the concepts of the 'general systems theory' (GST) of Ludwig von Bertalanffy to explain changes of the physical climate, in particular the causal relationship between 'iňトnite ice melting' (the cause M) and 'ice mass Losses' (The effect). The 'general systems theory' of Ludwig von Bertalanffy is a holistic, controversial theory, started in 1938, that may qualitatively explain some phenomena in ecology and social sciences but is far from being accepted as a science subjected to veriiňトcation and falsiiňトcation.

Author: The 'general systems theory' (GST) of Ludwig von Bertalanffy (1968) is not applied to explain the physical climate in the manuscript. It is mentioned in order to put the concept of finite cause and effect interaction (FCEI) in systemic framework of control system theory (CST). The revised article provides broader description of the

[Figure]

FCEI conceptual platform based on the principles of the control system theory (CST) (Wiener, 1948), of the natural causal process theories (nCPT) (Russell 1948; Salmon 1984), of the circular cumulative causation theory (CCT) (Myrdal, 1957), of the general system theory (GST) (von Bertalanffy, 1968) and particularly of the ideal feedback amplifier (e.g. Kuo and Farid, 2003).

Reviewer #1: Moreover, it presents a simplistic theory of the linear feedback which comes out as a very particular case of the much more general and well-grounded and mathematically funded 'Control system's theory'.

Author: Eqns. 3-5 in Sec. 2.2 follows from the basic relations of ideal feedback amplifier. The article for the purpose of this study redefines the common terms 'input' and 'output' of ideal feedback amplifier in terms 'cause C' and 'effect E' of continuous infinite causal processes (Fig. 1). The open loop gain is denoted in the article as propensity p, the closed loop gain is q and feedback factor is f. This model is not appropriate for finite causal processes. The mathematics of the finite continuous causal processes (Eq. 11-12 of Sec. 3) is not possible without the newly introduced decomposition of basic terms (Eqns. 3-5) into primary effect and interaction (Eq. 6). Decomposition enables the definition of the relation of the interaction term caused by effect of the proportional feedback and the residual causal capacity CU-C (Eq. 12, Fig. 2) instead to the elapsed cause C (Eq. 8, Fig. 1) as it is for infinite processes. The variable rate of sharing of the causal capacity C/(CU-C) between the elapsed cause C inducing feedback F(C) in the past and the remaining cause (CU-C) (12) in the future, characterizes the influence of the finiteness on the causal process or how permanently apply, take or use something more of something finite. From Eq. 12 directly follows by integration the term for interaction in Eq. 13, the term for overall effect Eq. 15 and the interaction potential Eq. 17. All expressions are easily calculable. The decomposition (Eq. 6) enlightens also some important relations. The two terms: interaction and feedback, colloquially often alternate. In Eq. 6, the article introduces a mathematical distinction between interaction and feedback necessary for understanding of these terms in the

manuscript. Interaction is defined as the contribution to the primary effect induced by the feedback affecting the primary cause with intensity i defined in Eq. 4 (Figs. 1 and 2). The article introduces the mathematical term Eq. 9 in Sec. 2.3, not used in CST for definition of theoretical interaction potential as a measure of how much efforts or works has to be done for progression of a complex continuous causal process. This term is important for assessments of interactions between complex causal processes and their environments. Appendix A brings forward a general term for derivatives of functions with respect to bounds of finite variables in a concise form based on infinitesimal considerations (Fig. A-1) on constant (Fig. A-2) and quadratic (Fig. A-3) causal relations. Examples of calculus in Appendix A confirm that the mathematical model of finite causal processes provides a unique facility for discovery and estimation of the final causal capacity of observed data. With all this respect, author believes that the mathematics in the article brings forward new appropriately formulated terms on the only possible and reasonable way. The mathematical model adequately describes the natural properties of finite causal processes in terms of CST useful for applications in system dynamics. It is a novel approach to the concept of finiteness derived by modification in terms of the control system theory and ideal feedback amplifier for study of general finite causal processes.

Reviewer #1: The application of GST formalism to the relationship between 'Finite ice Melting' and 'ice mass Losses' . . .

Author: The GST formalism is not applied to FMLI relation. The general finite cause (C) and effect (E) relation (FCE) with ultimate exhaustible finite causal capacity CU exposed in the article is not only philosophical or speculative concepts. In physical view particular cause C (e.g. ice melting M) and effect E(C) of C (e.g. ice mass losses L(M)) may be regarded as empirical physical laws of nature dominated by the finiteness of causal capacities CU (such as the total mass MU of ice sheets). The empirical idea of finiteness is a physical concept too but not only physical. FMLI is also a temporal concept since it relates observable physical processes, such as the

interaction between the ice melting M in the past and anticipated ice mass losses L(MU-M) due to melting of remaining mass of ice sheets MU-M in the future under changing environmental condition. Due to many complex hardly jointly manageable interrelated causes and effects, the study aims to find out if there are global synthetic measurable theoretically founded quantitative parameters of a general mathematical model appropriate for analysis of complex physical relations of climate change and interactions between ice melting and ice mass losses.

Reviewer #1: . . .seems therefore inappropriate giving rise to 'vague' concepts without any physical correspondence.

Author: The revised manuscript presents a more detailed description of the physical aspect of the FMLI model based on thermo-dynamical balance in renamed Sec.4.2 Thermal interaction of ice sheets and environment. The reported acceleration of ice mass losses L[M(T)] (the effect E) alarmed that an growing amount of heat (thermal energy) Q[M(T)] of the climate system (e.g. in GJ) has been transferring to the melting mass M(T) (the cause C) of ice sheets during the observation time T. After achieving the melting temperature and melting of some mass M(T) of ice due to permanent heat flow dQ/dT from environment heat to ice sheets in time T, the further lessening of residual ice mass MU-M(T) together with shrinkage of exposed area of ice sheets is continuing. During melting of surface ice at constant pressure, the temperature gradient of ice sheets supposedly remains approximately constant. Therefore the mean temperature of ice sheet remains nearly constant during the whole melting process in time T. Hence, the heat contents (the enthalpy H, e.g. in GJ) of the mass M(T) of melted ice is H[M(T)] and of the residual mass MU-M(T) of ice sheets is H[MU -M(T)]. The reported ice mass anomaly suggests that the environmental heat Q[M(T)] induces a feedback as in (7) in some proportion i to the heat content H[M(T)] of the mass of melted ice M(T) of the ice sheets in time T is F{Q[M(T)]}=iQ[M(T)]=iH[M(T)]. The mathematical rate of the finite interaction (12) in time T relates the thermodynamical effect of the growing environmental heat Q[M(T)] and the lessening residual heat content H[MU-M(T)] that

under before mentioned conditions and assumptions represents the interaction rate of the ice mass losses L[M(T) and ice melting M(T) is dIF[(L,M(T)]/dT=iQ[M(T)]/H[MU - M(T)]=iMt/[MU -M(T)]. Since the last term has the same form with respect to variable M as the interaction rate (12) with respect to C, the FCEI, the mathematical model (11-20) may be applied to FMLI problem. With all this respect, author thinks that the physics of finite causal processes is correctly formulated for the purpose of this manuscript and that it appropriately applies the thermo-dynamical relation of global warming and interaction between ice melting and ice mass losses in terms of heat transfer from the climate system and the enthalpy of to the ice sheets.

Reviewer #1: . . .Examples of that are quoted from the manuscript. 1) 'This study of finiteness of natural processes recognizes the temporal FCEI (Finite Cause-and-Effect Interaction) empirical concept as a continuous sharing of irreplaceable and restricted overall ultimate causal capacity CU between the observable elapsed effect E in the past and the imperceptible but conceivable forthcoming limited exhaustible cause C beyond the instant of observation in the future. 2) The trans-temporal finite interaction implies the empirical link in continuation of the known uninterrupted past and the imaginable but finite perpetuating future separated by the instant of observation at the present time' 3) The mathematical model of the FCEI in this study considers a simple intuitive term of the continuous residual causal capacity R(C)=CU-C after spending some primary effect E'(C) (1) of the limited cause C on the expense of the ultimate cause CU.

Author: The first two statements in the manuscript wordily explains the temporal character of the mathematical rate of change E(C)/(CU-C) in the FCEI definition (Eqn. 12) as the ratio between the growing elapsed effect E(C) in the past and the diminishing remaining driving cause (CU-C) with respect to the waste of ultimate causal capacity CU in the future separated by the instant of observation (Fig. 2). The third statement represents the conservation principle between elapsed effect and forthcoming cause which expresses the assumption that a continuous finite causal relation does not change its

properties of propensity to and intensity of interaction during unchanged environmental conditions with respect to wasting of ultimate causal capacity CU over time (Eqns. 11-20).

Reviewer #1: Beyond the above criticisms, the author tries to make millenary climatic predictions (extrapolations) using the simplistic GST relationships, as quoted from the manuscript: 'It is possible to predict by extrapolating the FMLI ice mass loss curve (28) that the melting out of the total mass of ce MU=2.50x106 Gt due to the interaction with climate change under same environmental conditions could happen in the year TM=285070 with 8% uncertainty of ultimate ice mass MU estimation (Figs. 3 and 4). That prediction is totally speculative and cannot be accepted.

Author: The mathematical model of finite causal processes provides analytic parameters based on observed data which allow extrapolations beyond the instant of observation. Following the reviewer's comment the long term predictions are omitted from the revised manuscript. Instead, short term extrapolations to the end of the 21th century are given.

Reviewer #1: Moreover, the author does totally ignore alternative approaches studying the causality in the climatic system (e.g. Granger causality) ...

Author: In the revised manuscript a comment is added why the Granger statistical causality is not used. Granger statistical causality investigates causal dependency between stochastic variables and do not indicate real causality in the deterministic context of the manuscript. Instead of Granger statistical concept of causality, the manuscript focuses on deterministic physical interpretations of work done by environmental effects and inherent physical properties of ice sheets giving the propensity, sensitivity and interaction intensity parameters of the mathematical model by derivatives and integration of the FCEI interaction curves (Eqns. 11-20 and Fig. 2). With respect to the scope and size of the manuscript no statistical analysis is planned.

Reviewer #1: ... and therefore it is not understood in which the manuscript adds new

knowledge.

Author: The manuscript presents a novel interdisciplinary approach to system dynamics that conceptualized, modelled and quantified the influence of finiteness of deterministic continuous causal processes with exhaustible causal capacities what was not applied earlier in investigations of Earth mechanisms. The manuscript introduces the finiteness as a physical property inseparable of other physical properties of continuous processes with limited causal capacities. It provides apt mathematical model of finite causal processes with reasonable numerical, geometrical and physical interpretations of the proposed concept. The calculus in the manuscript implies a new general definition of derivatives of functions with respect to bounds of finite variables. The study also uncovers a new application of the mathematical model that makes it possible to discover and to estimate the unknown ultimate causal capacities from relevant information of supposedly finite causal processes beyond the instant of observation. The proposed analytics based on thermo-dynamical balance confirms that the climate system and ice mass anomaly monitored recently on Greenland and Antarctica ice sheets under global interaction of a combination of various component systems, such as the atmosphere, cryosphere, hydrosphere, oceans and human activities may be viewed as finite dynamical causal processes by definition given in the manuscript. Extrapolations suggest dates of beginning of intensive ice melting and possible ice mass losses in the future. The reverse numerical calculation procedure satisfactorily re-estimated the total ice mass of ice sheets from observed data within the suggested limits. The numerical results are in almost perfect agreement with ice mass anomaly data observed on Greenland and Antarctica.

Reviewer #1: Giving the above arguments, the manuscript must be rejected and cannot be accepted to 'Earth System Dynamics' journal.

Author: *The introduction of finiteness in studies of natural phenomena is important for understanding and analysis of finite processes in earth system dynamics. There are many apparently finite causal processes which might be studied by using coherent mathematical apparatus capable to deal with finiteness. *Author's opinion is that the article brings forward a novel concept of finiteness in study of natural phenomena and original adequate mathematical model of dynamics of finite causal processes in terms of modified control system theory what is elaborated in more details in the revised manuscript. *Author also believes that the example 'Climate change and ice melting interaction' in the revised manuscript is properly physically elaborated in terms of thermo-dynamical balance, the underlying concept of finiteness and applied mathematical model of finite causal processes. The results of the numerical calculations are in agreement with systematic collection of observations on ice melting and ice mass losses on Greenland and Antarctica. *The manuscript fulfils in many aspects the declared aims and scopes of the ESD journal. *For all this reasons author thinks that the submitted and revised manuscript should be considered for publishing in order to add impetus to further investigation of finite causal processes of interests in earth system dynamics and elsewhere.

---

## Author Comment (AC4) · 7 Aug 2018

Reviewer #2: I strongly sympathize with the research questions that motivated this study, because they are extremely relevant and timely in general physics and in earth system dynamics in particular. The author takes a bold approach to causality that, at first sight, intuitively makes a lot of sense in view of open philosophical discussions that have been going on for a long time.

Submission of this article to ESD was encouraged by enthusiastic aims and scopes of this journal.

Reviewer #2: In my opinion, the fundamental problems are that the theoretical bases are far from being consolidated and the mathematical formulation is far from being fully matured.

[Figure]

Author: Eqns. 3-5 in Sec. 2.2 follows from the basic relations of ideal feedback amplifier. The article redefines the common terms 'input' and 'output' of ideal feedback amplifier in terms 'cause C' and 'effect E' of continuous infinite causal processes (Fig. 1). The open loop gain is denoted in the article as propensity p, the closed loop gain is q and feedback factor is f. However, this model is not adequate for finite causal processes.

The mathematics of the finite continuous causal processes (Eq. 11-12 of Sec. 3) is not possible without the newly introduced decomposition of basic terms (Eqns. 3-5) into primary effect and interaction (Eq. 6). Decomposition enables the definition of the relation of the interaction term caused by effect of the proportional feedback and the residual causal capacity CU-C (Eq. 12, Fig. 2) instead to the elapsed cause C (Eq. 8, Fig. 1) as it is for infinite processes. The variable rate of sharing of the causal capacity C/(CU-C) at the moment of observation between the elapsed cause C inducing feedback F(C) in the past and the remaining cause (CU-C) (12) in the future, characterizes the influence of the causal finiteness on the causal process or how permanently apply, take or use something physically more of something physically finite. From Eq. 12 directly follows by integration the term for interaction in Eq. 13, the term for overall effect Eq. 15 and the interaction potential Eq. 17. All expressions are easily calculable.

The decomposition (Eq. 6) enlightens also some important features. The two terms: interaction and feedback, colloquially often alternate. In Eq. 6, the article introduces a mathematical distinction between interaction and feedback necessary for understanding of these terms in the manuscript. Interaction is defined as the contribution to the primary effect induced by the feedback affecting the primary cause with intensity i defined in Eq. 4 (Figs. 1 and 2).

The article introduces the mathematical term Eq. 9 in Sec. 2.3, not used in CST for definition of theoretical interaction potential as a measure of how much efforts or works has to be done for progression of a complex continuous causal process. This term

is important for assessments of interactions between complex causal processes and their environments.

Appendix A brings forward a general term for derivatives of functions with respect to bounds of finite variables in a concise form based on infinitesimal considerations (Fig. A-1) on constant (Fig. A-2) and quadratic (Fig. A-3) causal relations. Examples of calculus in Appendix A confirm that the mathematical model of finite causal processes provides a unique facility for discovery and estimation of the final causal capacity of observed data.

With all this respect, author believes that the mathematics in the article brings forward new appropriately formulated terms on the only possible and reasonable way. The mathematical model adequately describes the properties of finite causal processes in modified terms of CST useful for applications in system dynamics. It is a novel approach to the concept of finiteness in terms of the control system theory and ideal feedback amplifier for study of general finite causal processes.

Reviewer #2: I cannot recommend this manuscript for publication due to fundamental mathematical and physical concerns (which the other reviewer detailed carefully), especially the unproven validity of the theories and unproven physical reasonability of the formulations.

Author: The finite cause (C) and effect (E) relation (FCE) with ultimate exhaustible finite causal capacity CU exposed in the article are not only philosophical or speculative concepts. The empirical idea of finiteness is also a physical concept per se but not only physical. In physical view particular cause C (e.g. ice melting M) and effect E(C) of C (e.g. ice mass losses L(M)) may be regarded as empirical physical laws of nature dominated by the finiteness of causal capacities CU (such as the total mass MU of ice sheets).

It is also a temporal concept since it relates observable physical processes, such as the interaction between the ice melting M in the past and anticipated ice mass losses

L(MU-M) due to melting of remaining mass of ice sheets MU-M in the future under changing environmental condition.

Due to many complex hardly jointly manageable interrelated causes and effects, this study aims to find out if there are global synthetic measurable theoretically founded quantitative parameters of a general mathematical model appropriate for analysis of complex physical relations of climate change and interactions between ice melting and ice mass losses.

The revised manuscript presents a more detailed description of the physical aspect of the FMLI model based on thermo-dynamical balance in renamed Sec.4.2 Thermal interaction of ice sheets and environment.

The reported acceleration of ice mass losses L[M(T)] (the effect E) alarmed that an growing amount of heat (thermal energy) Q[M(T)] of the climate system (e.g. in GJ) has been transferring to the melting mass M(T) (the cause C) of ice sheets during the observation time T. After achieving the melting temperature and melting of some mass M(T) of ice due to permanent heat flow dQ/dT from environment heat to ice sheets in time T, the further lessening of residual ice mass MU-M(T) together with shrinkage of exposed area of ice sheets is continuing. During melting of surface ice at constant pressure, the temperature gradient of ice sheets supposedly remains approximately constant. Therefore the mean temperature of ice sheet remains nearly constant during the whole melting process in time T. Hence, the heat contents (the enthalpy H, e.g. in GJ) of the mass M(T) of melted ice is H[M(T)] and of the residual mass MU-M(T) of ice sheets is H[MU -M(T)].

The reported ice mass anomaly suggests that the environmental heat Q[M(T)] induces a feedback as in (7) in some proportion i to the heat content H[M(T)] of the mass of melted ice M(T) of the ice sheets in time T is F{Q[M(T)]}=iQ[M(T)]=iH[M(T)].

The mathematical rate of the finite interaction (12) in time T relates the thermodynamical effect of the growing environmental heat Q[M(T)] and the lessening residual heat

content H[MU-M(T)] that under before mentioned conditions and assumptions repre­sents the interaction rate of the ice mass losses L[M(T)ïAÌ and ice melting M(T) is dIF[(L,M(T)]/dT=iQ[M(T)]/H[MU -M(T)]=iMt/[MU -M(T)].

Since the last term has the same form with respect to variable M as the interaction rate (12) with respect to variable C of the FCEI, the mathematical model (11-20) may be applied to FMLI problem.

With all this respect, author thinks that the physics of finite causal processes in the article is correctly formulated and that it appropriately explains the relation of global warming and interaction between ice melting and ice mass losses in terms of heat transfer from the climate system and the enthalpy of the ice sheets.

The introduction of finiteness in studies of natural phenomena is important for under­standing and analysis of finite processes in earth system dynamics. There are many apparently finite causal processes which might be studied by using coherent mathe­matical apparatus capable to deal with finiteness.

*Author's opinion is that the article brings forward a novel concept of finiteness in study of natural phenomena and original adequate mathematical model of dynamics of finite causal processes in terms of modified control system theory what is elaborated in more details in the revised manuscript.

*Author also believes that the example 'Climate change and ice melting interaction' in the revised manuscript is properly physically elaborated in terms of thermo-dynamical balance, the underlying concept of finiteness and applied mathematical model of finite causal processes. The results of the numerical calculations are in agreement with systematic collection of observations on ice melting and ice mass losses on Greenland and Antarctica.

*The manuscript fulfils in many aspects the declared aims and scopes of the ESD journal.

[Figure]

*For all this reasons author thinks that the submitted and revised manuscript should be considerd for publishing in order to add impetus to further investigation of finite causal processes of interests in earth system dynamics and elsewhere.